# Association between vision impairment and mortality: protocol for a systematic review and meta-analysis

Joshua R Ehrlich [1,2] Jacqueline Ramke,[3,4] David Macleod,[4,5] Bonnielin K Swenor,[6,7] Helen Burn,[8] Chan Ning Lee,[4,9] William J Waldock,[10] Justine H Zhang ,[4,11] Iris Gordon,[4] Nathan Congdon,[12,13] Matthew Burton,[4,14] Jennifer R Evans[4]

For numbered affiliations see end of article.

**Correspondence to**
Dr Joshua R Ehrlich;
joshre@med.umich.edu

## ABSTRACT

**Introduction** Due to growth and ageing of the world's population, the number of individuals worldwide with vision impairment (VI) and blindness is projected to increase rapidly over the coming decades. VI and blindness are an important cause of years lived with disability. However, the association of VI and blindness with mortality, including the risk of bias in published studies and certainty of the evidence, has not been adequately studied in an up-to-date systematic review and meta-analysis.

**Methods and analysis** The planned systematic review and meta-analysis will adhere to the Preferred Reporting Items for Systematic Reviews and Meta-Analyses guidelines. Databases, including MEDLINE Ovid, Embase Ovid and Global Health, will be searched for relevant studies. Two reviewers will then screen studies and review full texts to identify studies for inclusion. Data extraction will be performed, and for included studies, the risk of bias and certainty of the evidence will be assessed using the Grades of Recommendation, Assessment, Development and Evaluation approach. The prognostic factor in this study is visual function, which must have been measured using a standard objective ophthalmic clinical or research instrument. We will use standard criteria from WHO to categorise VI and blindness. All-cause mortality may be assessed by any method one or more years after baseline assessment of vision. Results from included studies will be meta-analysed according to relevant sections of the Meta-analysis Of Observational Studies in Epidemiology checklist.

**Ethics and dissemination** This review will only include published data; therefore, ethics approval will not be sought. The findings of this review and meta-analysis will be published in an open-access, peer-reviewed journal and will be included in the ongoing *Lancet Global Health* Commission on Global Eye Health.

## Strengths and limitations of this study

► This is an up-to-date systematic review and meta-analysis to determine the nature and extent of published literature on the association of vision impairment and blindness with mortality.
► This review will comprehensively assess published peer-reviewed English-language manuscripts, with no time period or geographical restrictions.
► This will be the first review to carry out a formal assessment of risk of bias in included studies and the certainty of the evidence on this topic using the Grades of Recommendation, Assessment, Development and Evaluation approach.
► A potential limitation might be the paucity of published literature on how specific levels of vision impairment contribute to mortality.
► Another potential limitation is that the complexity of pathways between eye health and mortality is unlikely to be fully described and tested in the current literature.

that individuals with vision impairment (VI) have an increased risk of mortality compared with those with normal vision.[3] However, an up-to-date systematic review and meta-analysis of the published literature, including a formal assessment of risk of bias and certainty of the evidence, is needed to characterise the relationship between VI and mortality globally.

In order to guide a systematic review and meta-analysis of the association of VI with mortality, we developed a theoretical framework adapted from the WHO International Classification of Functioning.[4] Our framework illustrates the possible relationship between VI and mortality, as well as the diverse mediating and moderating factors that may contribute to this association (figure 1). As depicted, we hypothesise that VI, operationalised as a decline in visual function, is associated with mortality through its effects on systemic health (eg, an increased risk of chronic disease, frailty

## INTRODUCTION

More than 250 million people globally are blind or visually impaired, and the number affected is projected to increase due to growth and ageing of the world's population.[1] Poor vision is associated with an increased risk of dementia, depression, falls and loss of independence.[1 2] Some prior studies have also reported

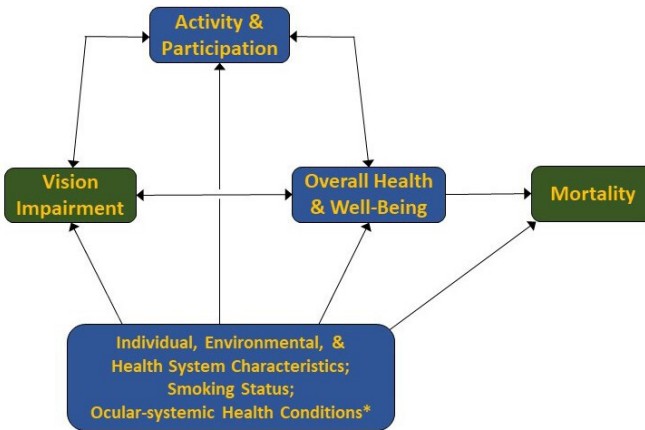

**Figure 1** Theoretical framework for the association of vision impairment and mortality. This figure presents the hypothesised relationships between vision impairment and mortality that inform the systematic review and meta-analysis. *Conditions such as cardiovascular disease, diabetes, hypertension and stroke that increase the risk of both vision impairment and mortality. Adapted from the model of disability in international classification of functioning, disability and health:ICF. Geneva:WHO;2001. ICF, International Classification of Functioning.

and decreased functional status). Factors such as participation (social, physical, daily activities) both impact and are impacted by visual and systemic health (eg, VI increases the risk of social isolation, which in turn affects overall health). Finally, individual-level traits, environmental and health system characteristics, smoking and conditions with both ocular and systemic manifestations (eg, cardiovascular disease, diabetes, hypertension, stroke) may simultaneously increase the risk of both VI and mortality.

A prior systematic review and meta-analysis published in 2016 summarised findings from 29 prospective studies that assessed the association between vision and the risk of mortality.[3] That study reported that the risk of death was 36% higher in the group with the highest level of VI compared with those without VI, and that for each 0.1 increment change in logarithm of minimum angle of resolution (logMAR) the risk of death increased by 4%. However, the study had several limitations. First, it did not assess or account for the level of bias in included studies or certainty of the evidence. Additionally, three included studies assessed VI based on billing codes, and seven used self-reported VI rather than objective quantifiable measures. Self-reported visual function may reflect a distinct latent construct,[5] in which case the pooled analysis of studies that assessed visual function objectively and subjectively may bias results in an unpredictable fashion. The highest level of VI was used as a predictor of mortality even though VI categories varied from study to study. Search terms also did not include specific eye conditions (eg, glaucoma or cataract), so studies of the association between mortality and VI due to these conditions may have been omitted. Finally, several prospective studies have been published in recent years that report the association between VI and mortality

in geographic regions, such as sub-Saharan Africa, that were under-represented in the prior systematic review and meta-analysis.

The study described in this protocol will seek to provide an updated review of the literature and estimate of the effect of VI on the risk of mortality. Notably, the complex pathways that may mediate the association between VI and mortality may not have been fully described or tested in prior studies, though doing so will be an important future step toward optimising outcomes for those with VI and blindness. By including an assessment of the risk of bias to inform an overall judgement of the certainty of the evidence, and by considering newly published studies from under-represented geographical regions, this systematic review and meta-analysis will make an important contribution to global eye health.

### Objectives/review questions
This systematic review and meta-analysis will aim to answer the following questions:
1. What is the extent, strength and quality of the published evidence that VI is associated with the risk of all-cause mortality?
2. To what degree does VI affect the risk of all-cause mortality, and does this risk vary based on level of visual function?

## METHODS AND ANALYSIS
### Protocol and registration
This work was undertaken as part of the *Lancet Global Health* Commission on Global Eye Health. We have drawn on the PROGnosis RESearch Strategy framework for prognosis research in developing this protocol.[6] The protocol has been registered prospectively with the Open Science Framework (OSF) registry and can be viewed at: http://osf.io/weu96. Any future amendments to the protocol will be noted in the OSF registration. Results of the systematic review and meta-analysis described herein will be reported according to the relevant section of the Preferred Reporting Items for Systematic Reviewa and Meta-Analyses (PRISMA) checklist for systematic reviews.[7] The completed PRISMA-Protocols checklist is presented in online supplementary appendix 1.[8]

### Search method for identification of studies
#### Electronic searches
We will search the following electronic databases:
1. MEDLINE Ovid (1946–2020).
2. Embase Ovid (1980–2019).
3. Global Health (1973–2020).

The full electronic search strategy for MEDLINE Ovid is included in online supplementary appendix 2. We will not limit the search by date. Articles will be included if they were published before 1 February 2020. As noted above, the search will be limited to English language articles and will not include conference abstracts or grey literature.

### Searching other resources

We will identify additional studies by searching the reference lists of relevant publications identified through the electronic searches and by searching any prior review articles on this topic.

### Criteria for considering studies to review
#### Types of studies

We will include published prospective and retrospective cohort studies with a baseline assessment of the exposure (vision) and longitudinal assessment of the outcome (all-cause mortality) over a period of at least 1 year. Since age is a strong risk factor for mortality and VI, estimates of the effect of VI on mortality risk must be age adjusted. Interventional studies and studies where all participants had a specific systemic disease (eg, diabetes) will be excluded due to the difficulty of separating the possible effect of VI on mortality from the effect of an intervention or systemic disease on mortality. Only peer-reviewed articles published in English will be included. We will not include grey literature or conference abstracts. We will consider publications from all years and geographical regions.

#### Types of participants

Men and women aged 40 years and above at the time of enrolment will be eligible for inclusion. We are restricting the population to this age group of because of the low rate of mortality in younger individuals.

#### Types of prognostic factors

The prognostic factor in this study is visual function. Visual function must have been measured using a standard objective ophthalmic clinical or research instrument, including, but not limited to visual acuity, visual fields, contrast sensitivity and stereoacuity. If a study contains data on the effect of multiple measures of visual function (eg, visual acuity and visual fields), we will report each of these and they will be included in the meta-analysis. Studies where visual function was self-reported or determined based on billing codes will not be included.

We will consider several cut-points:

1. For visual acuity, we will consider presenting binocular visual acuity or visual acuity in the better-seeing eye (if both are reported we will use the binocular measurement). Definitions of VI will be based on the categories of VI in the WHO International Classification of Diseases.[9] People with each of the following categories of VI will be compared with those with better vision.
   a. Mild VI or worse will be defined as visual acuity <6/12.
   b. Moderate VI or worse will be defined as visual acuity <6/18.
   c. Severe VI or worse will be defined as visual acuity <6/60.
   d. Blindness will be defined as visual acuity <3/60.
2. For other measures of vision, we will adopt study-specific definitions of VI since standardised definitions do not exist or are not widely used.

We will use the definitions of VI and blindness reported in the included studies that most closely correspond to these definitions. However, we anticipate heterogeneity in these measures across studies. For example, studies may vary in whether they consider uncorrected, presenting or best-corrected visual acuity; they may include measures of visual fields, stereoacuity and/or contrast sensitivity in their definition; they may consider the worse-seeing eye, better-seeing eye or binocular visual acuity; and they may employ different categorical definitions of VI and blindness. We will be inclusive but will explore this heterogeneity in meta-regression analyses (see Meta-regression section). When available, we will also use continuous measures of vision (eg, logMAR or log contrast sensitivity) in our analyses.

#### Types of outcome measures
*Outcome*

1. The outcome is all-cause mortality one or more years after baseline assessment of vision. Mortality may be reported using different measures of effect size and we will include all measures.

Ascertainment of death may be made by any method, including but not limited to review of vital records (eg, death certificates) or report by an informant. We have chosen to include studies that ascertained death by informant report since not all countries provide access to complete vital records and we seek to include studies from all regions of the world. If sufficient data are available, we will consider performing analyses to determine the association between VI and cause-specific mortality.

#### Measures of effect

To determine the association between VI and mortality, we will extract age-adjusted measures of effect reported in each included study. All measures of effect must be age adjusted. When available, we will extract measures that have adjusted for other theoretical confounders (eg, socioeconomic status, smoking status, cardiovascular disease, diabetes, hypertension, stroke) as depicted in our conceptual framework (figure). To the extent that it is possible to do so, we will choose measures of effect that are not adjusted for likely mediators on the pathway between VI and mortality, such as overall health or functional status.

### Data collection and analysis
#### Selection of studies

Two review authors will independently screen search results based on title and abstract and will remove reports that clearly do not fall into the scope of this review. Disagreements will be resolved by discussion and consultation with another author as needed. We will acquire the full text of all publications appearing to potentially meet criteria for inclusion in this review. Two review authors will screen all of these reports for method of visual function assessment, type of study design, duration of follow-up, and ascertainment of death. Any disagreements will be discussed and if they cannot be resolved will be arbitrated by a third review author. Screening of search results will be conducted using

Covidence systematic review software (Veritas Health Innovation, Melbourne, Australia; available at www.covidence. org).

## Data extraction and management

Data extraction will be guided by the relevant sections of the CHecklist for critical Appraisal and data extraction for Systematic Reviews of prediction Modelling Studies checklist.[10] Two review authors will independently extract the following data from each included study: study design, participant characteristics, study population and size, study setting, study dates, follow-up duration, diagnostic and ascertainment methods, study attrition, estimates of effect size, and standard errors. Disagreements will be resolved by discussion and consultation with another author as needed; if they cannot be resolved they will be arbitrated by a third review author. The types of data likely to be reported as estimates of effect size include hazard ratios, risk ratios, rate ratios, standardised mortality ratios, cumulative incidence rates, proportions, survival curves and/or ORs. Data extraction and management will be conducted using Covidence systematic review software.

## Assessment of risk of bias

Two reviewers will independently assess the risk of bias in each included study using the Quality in Prognostic Studies (QUIPS) tool.[11] They will assess study participation, attrition, prognostic factor measurement, outcome measurement, confounding and statistical analysis and reporting. Likely confounders include systemic health conditions that increase risk of VI and mortality (eg, diabetes), access to medical care, socioeconomic status and smoking status. For each QUIPS domain, we will assign a rating of low, moderate or high risk of bias. Ratings of each of the domains in QUIPS will be considered to provide an overall risk of bias assessment for each study. Only studies receiving a rating of low risk of bias in all of the aforementioned domains will be given an overall 'low' rating; any study that received one or more ratings of high risk of bias will receive an overall 'high' rating; other studies will receive an overall 'medium' risk of bias rating.

## Measures of association

We will extract summary measures of the association between VI and risk of mortality. We anticipate that some studies will report an overall event rate for the study period, while others may provide estimates of effect size. For all estimates, we will extract SEs; if they are not reported we will extract 95% CIs and use these to calculate standard errors. As noted, we will preference measures that are adjusted for theoretical confounders but not mediators of the association between VI and mortality. We will extract definitions of VI and blindness to permit analyses based on specific levels of VI or blindness, insofar as there are sufficient data available to do so.

## Dealing with missing data

We will include studies that follow individuals with and without VI (or with varying levels of VI) for 1 or more years and report the proportion who died, even if there are missing data. If all of the necessary information are not found in a published study, for articles published in 2010 or later we will email the corresponding author to solicit further information. If we are unable to obtain the necessary information, we will document in the review that we attempted to contact the study authors. We will consider the sensitivity of our meta-analysis to the effect of missing data. We will analyse the data that is available rather than imputing missing data. We will document and discuss the possible effect of missing data on each study and on the overall review and meta-analysis.

## Assessment of heterogeneity

Clinical heterogeneity will be assessed by comparing key participant characteristics at the study level (eg, age, sex, ocular diagnoses). Methodological heterogeneity will also be considered, including a comparison of the risk of bias of included studies. We will assess statistical heterogeneity by inspecting forest plots and through inspection of the $I^2$ and $\tau^2$ statistics to examine the proportion of heterogeneity across studies that is due to chance. If high levels of heterogeneity are detected ($I^2 > 50\%$), we will explore likely sources of this heterogeneity (see the Meta-regression section). We will also assess small study effects, one of which may be publication bias, by preparing a funnel plot,[12] which is a scatter plot of effect size versus precision (SE).

## Data synthesis

### Data synthesis and meta-analysis approaches

Methods and results of our meta-analysis will be guided by relevant sections of the Meta-analysis Of Observational Studies in Epidemiology checklist.[13] Meta-analyses will be performed using a random-effects, generic inverse variance meta-analysis model in Stata V.16 (StataCorp). Random-effects, rather than fixed effects, models will be used since it is likely that the true effect of VI on mortality varies from study to study due to differences in study populations and contexts. The meta-analysis will be summarised using the pooled estimate, its 95% CI, and between study variance ($\tau^2$). The meta-analysis will be performed and results will be reported for adjusted effect estimates. We will conduct meta-analyses separately for the different types of effect measures (eg, HRs, ORs and risk ratios). The log of each study estimate and its confidence intervals will be used to determine the study SE; these will be then pooled using random-effects meta-analysis before taking the exponent of the results to present the pooled effect estimate on the original scale. We will assess and report the overall quality of evidence from our meta-analysis using the modified Grades of Recommendation, Assessment, Development and Evaluation tool.[14]

### Meta-regression

Where data permit, we will investigate the relationship between the following covariates and effect size using random-effects meta-regression:

▶ Sex,

- ► Average age,
- ► Method(s) used to measure visual function (visual acuity, visual field),
- ► Duration of follow-up,
- ► Global super-region as defined in the Global Burden of Disease study.[15]

The meta-regression outcome variable will be the log of the effect estimate for each study, and the aforementioned covariates will be included where data are available to do so.

## Sensitivity analyses

We will conduct a sensitivity analysis in which studies are excluded if they are judged to be at high risk of bias.

## Patient and public involvement statement

As we plan to review existing published literature only, this review will be performed without specific patient or public involvement.

## ETHICS AND DISSEMINATION

Ethics approval is not required, as our review will only include published data. Findings will be published in an open-access peer-reviewed journal and a summary of results will also be included in the ongoing *Lancet Global Health* Commission on Global Eye Health.[16] We anticipate that the findings will be of considerable interest to those involved in eye health provision, as well as the general medical, public health, development and governmental sectors.

## Author affiliations

[1]Ophthalmology and Visual Sciences, University of Michigan, Ann Arbor, Michigan, USA
[2]Institute for Healthcare Policy and Innovation, University of Michigan, Ann Arbor, Michigan, USA
[3]School of Optometry and Vision Science, The University of Auckland, Auckland, New Zealand
[4]International Centre for Eye Health, London School of Hygiene and Tropical Medicine, London, UK
[5]MRC Tropical Epidemiology Group, London School of Hygiene and Tropical Medicine, London, UK
[6]Ophthalmology, Johns Hopkins University, Baltimore, Maryland, USA
[7]Epidemiology, Johns Hopkins University, Baltimore, Maryland, USA
[8]Ophthalmology Department, Stoke Mandeville Hospital, Aylesbury, UK
[9]St. Paul's Eye Unit, Royal Liverpool University Hospital, Liverpool, UK
[10]School of Clinical Medicine, University of Cambridge, Cambridge, UK
[11]Manchester Royal Eye Hospital, Manchester, UK
[12]Global Eye Health, Queen's University Belfast, Belfast, UK
[13]Zhongshan Ophthalmic Center, Sun Yat-Sen University, Guangzhou, Guangdong, China
[14]Moorfields Eye Hospital, London, UK

**Contributors** All authors made substantive intellectual contributions to the development of this protocol. JRE, JR, MB and JE conceptualised the review approach. JRE drafted the first version. JR, MB and JE provided guidance to the research team. DM, BS, IG, NC, HB, CNL, WJW and JHZ developed the draft further. All authors were involved in revisions of the manuscript, developing review questions and the review design. All authors approved the final version of the manuscript. JRE and JE took overall responsibility for the content of this manuscript.

**Funding** MB is supported by grants from the Wellcome Trust (207472/Z/17/Z) to MB. JR is a Commonwealth Rutherford Fellow, funded by the UK government through the Commonwealth Scholarship Commission in the UK. BKS is supported by the National Institutes of Health (K01AG052640). JRE is supported by the National Institutes of Health (K23EY027848). The Lancet Global Health Commission on Global Eye Health is supported by The Queen Elizabeth Diamond Jubilee Trust, Moorfields Eye Charity (GR001061), NIHR Moorfields Biomedical Research Centre, The Wellcome Trust, Sightsavers, The Fred Hollows Foundation, The SEVA Foundation, The British Council for the Prevention of Blindness and Christian Blind Mission.

**Disclaimer** No funder had any role in the design or conduct of this work.

**Competing interests** None declared.

**Patient and public involvement** Patients and/or the public were not involved in the design, or conduct, or reporting, or dissemination plans of this research.

**Patient consent for publication** Not required.

**Provenance and peer review** Not commissioned; externally peer reviewed.

**ORCID iDs**
Joshua R Ehrlich http://orcid.org/0000-0002-0607-3564
Justine H Zhang http://orcid.org/0000-0001-8385-2003

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
