## [Reviewer comments · BMJ Open]

ARTICLE DETAILS

TITLE (PROVISIONAL)	The Association Between Vision Impairment and Mortality: Protocol for a Systematic Review and Meta-Analysis
AUTHORS	Ehrlich, Joshua; Ramke, Jacqueline; Macleod, David; Swenor, B; Burn, Helen; Lee, Chan; Waldock, William; Zhang, Justine; Gordon, Iris; Congdon, Nathan; Burton, Matthew J; Evans, Jennifer

VERSION 1 – REVIEW

REVIEWER	Ryo Kawasaki Osaka University, Japan
REVIEW RETURNED	17-Mar-2020

GENERAL COMMENTS	This is a protocol paper for the systematic review and meta-analysis on the association between vision impairment and mortality. The protocol proposed is following the standardized reporting format of the PRISMA, using GRADE approach, and MOOSE, which is fine. Major comments: 1) Abstract should describe more theme specific information. It is somewhat not informative just to state that it will follow the standards or guidelines for reporting because they are minimum requirements. Please consider to provide more theme specific issues, such as definitions of VI/blindness, and how to confirm the mortality outcome etc..2) Will there be any possibility of competing risk especially for the diabetic patients or older age group for cardiovascular or cancer etc.? How do authors cope with this?3) I am sure authors conducted a preliminary literature search based on the defined strategy in Appendix 2. Were there enough studies to conduct the planned analysis? I think it might be helpful to provide a prelim search results to confirm that this design is feasible. Minor comments and suggestions: 1) Will this study involve collecting refractive status? Presenting VA measured at distance might not be equivalent between the myopic persons and hyperopic persons. It might be informative to stratify by refractive status.2) Will cause specific mortality be used in this study? Please consider this at least as a sub-analysis? This will provide insights into why VI/blindness are linked with higher mortality.
---

REVIEWER	Mingxing Wu State Key Laboratory of Ophthalmology, Zhongshan Ophthalmic Center, Sun Yat-sen University, Guangzhou, China
REVIEW RETURNED	04-Apr-2020

GENERAL COMMENTS	The authors present a protocol for a systematic review and meta-analysis to evaluate the association between vision impairment and mortality. Vision impairment and blindness are two important causes of mortality, and many published studies have provided some evidence, but the findings are not consistent. Such a systematic review and meta-analysis will provide a useful reference for global eye health. Overall, this protocol will need major revisions prior to consideration for publication. The following points should be addressed:  1. For "Types of participants", men and women aged 40 years or older were included for this protocol (Page 7 Lines 52-53), but for younger adults (age \leq 39 years, and \geq 16 years), such as ocular trauma and high myopia, was a high risk factor for vision impairment, which was also associated with mortality. I suggest the participants included them. 2. Vision impairment (VI) and blindness are two different states of visual dysfunction (see the definition by WHO), but mixed by the authors (Page 8 Lines 6-20). I suggest divide them into two parts, and analyze them respectively. 3. For "Electronic searches", no specific deadline was presented clearly, only "to present" (Page 9 Lines 7-13). 4. The statistical methods used in the protocol were not described fully. 5. For some systemic diseases, such as diabetes, they both cause vision impairment and mortality. In this case, how to evaluate the association between vision impairment and mortality was not presented clearly in this protocol.
---

REVIEWER	Marie-Josée Aubin Université de Montréal, Canada
REVIEW RETURNED	06-Apr-2020

GENERAL COMMENTS	Excellent. Important work. Well written. Comments:  1. One of the potential limitations identified (p.3/17) relates to the "complexity of pathways between eye health and mortality (...) [being] unlikely to be fully described and tested in the current literature." Could you elaborate on how you plan to address this difficulty, and the impact this could have on the theoretical framework presented (p. 6, 15/17)? 2. Sensitivity analysis (p. 12/17): "studies are excluded (...) if the risk of bias could not be adequately assessed". Would it be possible to describe (p. 10/17) the circumstances under which you expect this difficulty to occur?
--

REVIEWER	Liv Berit Augestad Departemnet of Neuromedicine and Movement Science. Faculty of Medicine and Health Sciece. Norwegian University of Science and Technology (NTNU).
REVIEW RETURNED	12-Apr-2020

GENERAL COMMENTS	This systematic review and Meta-analysis has the potential to make an important contribution to global eye health. It is a well written protocol.
---

	Some comments: 1) The authors have decided to exclude intervention studies where all participants had a specific systemic disease (e.g diabetes). 2) In addition, they have decided that men and women age 40 years and above at the time of the enrolment will be eligible for inclusion. I'm looking forward to read the strength and limitations (specific for 1 and 2) in the discussion chapter for the inclusion and the exclusion criteria.
REVIEWER	Rajeev Ramchandran Flaum Eye Institute, URMIC, University of Rochester, USA
REVIEW RETURNED	15-Apr-2020
GENERAL COMMENTS	A very well planned study using established, published methodology. The results will be be very meaningful for advancing the important of visual function in older adults.

VERSION 1 – AUTHOR RESPONSE

Reviewer: 1

This is a protocol paper for the systematic review and meta-analysis on the association between vision impairment and mortality. The protocol proposed is following the standardized reporting format of the PRISMA, using GRADE approach, and MOOSE, which is fine.

Major comments:

1) Abstract should describe more theme specific information. It is somewhat not informative just to state that it will follow the standards or guidelines for reporting because they are minimum requirements. Please consider to provide more theme specific issues, such as definitions of VI/blindness, and how to confirm the mortality outcome etc..

RESPONSE:

Thank you for this helpful comment. We have provided the suggested information in the abstract, as follows:

“The prognostic factor in this study is visual function, which must have been measured using a standard objective ophthalmic clinical or research instrument. We will use standard criteria from the World Health Organization to categorise VI and blindness. All-cause mortality may be assessed by any method one or more years after baseline assessment of vision.”

2) Will there be any possibility of competing risk especially for the diabetic patients or older age group for cardiovascular or cancer etc.? How do authors cope with this?

RESPONSE:

We agree that this is an important concern. This is the reason that we chose to exclude all studies that focused on specific populations (e.g. diabetes and stroke are common competing risk factors). We have also edited our conceptual framework in Figure 1 to explicitly point out that systemic health conditions like diabetes, hypertension, cardiovascular disease, and stroke are associated with both vision impairment and mortality. Based on this framework we are preferentially including effect estimates that adjust for likely confounders such as the presence of conditions like the aforementioned health conditions. Additionally, older age is likely the most common competing risk factor for mortality. Thus, we are only including studies that provide age-adjusted mortality estimates. Accordingly, we have noted on Page 6 (line 135), “Since age is a strong risk factor for mortality and VI, estimates of the effect of VI on mortality risk must be age-adjusted. Interventional studies and studies where all participants had a specific systemic disease (e.g. diabetes) will be excluded due to the difficulty of separating the possible effect of VI on mortality from the effect of an intervention on f... systemic disease on mortality”

3) I am sure authors conducted a preliminary literature search based on the defined strategy in Appendix 2. Were there enough studies to conduct the planned analysis? I think it might be helpful to provide a prelim search results to confirm that this design is feasible.

RESPONSE:

While a preliminary search was conducted, feasibility was also known based on the prior systematic review that we cite in the introduction.

Minor comments and suggestions:

1) Will this study involve collecting refractive status? Presenting VA measured at distance might not be equivalent between the myopic persons and hyperopic persons. It might be informative to stratify by refractive status.

RESPONSE:

We agree that this is an interesting and possible relevant consideration. However, very few epidemiologic studies that report VI and mortality include this type of detailed data on refractive status. While this will not likely be possible in the meta-analysis, we will consider noting this as a limitation when we report our findings.

2) Will cause specific mortality be used in this study? Please consider this at least as a sub-analysis? This will provide insights into why VI/blindness are linked with higher mortality.

RESPONSE:

As noted on Page 7, the outcome we will consider is all-cause mortality. We have added a statement on Page 7 (line 178) that, "If sufficient data are available, we will consider performing analyses to determine the association between VI and cause-specific mortality."

Reviewer: 2

Reviewer Name: Mingxing Wu

Institution and Country: State Key Laboratory of Ophthalmology, Zhongshan Ophthalmic Center, Sun Yat-sen University, Guangzhou, China Please state any competing interests or state 'None declared':
None declared

The authors present a protocol for a systematic review and meta-analysis to evaluate the association between vision impairment and mortality. Vision impairment and blindness are two important causes of mortality, and many published studies have provided some evidence, but the findings are not consistent. Such a systematic review and meta-analysis will provide a useful reference for global eye health. Overall, this protocol will need major revisions prior to consideration for publication.

The following points should be addressed:

1. For "Types of participants", men and women aged 40 years or older were included for this protocol (Page 7 Lines 52-53), but for younger adults (age ≤ 39 years, and ≥ 16 years), such as ocular trauma and high myopia, was a high risk factor for vision impairment, which was also associated with mortality. I suggest the participants included them.

RESPONSE:

We appreciate this insightful comment. It is true that participants younger than age 40 may have a higher risk of mortality associated with VI. However, analysis of the under 40 population is beyond the scope of the current study since both the causes of VI and mortality tend to be distinct in this group. Additionally, since the set of factors that both confound and mediate the VI-mortality relationship are likely different in the under and over 40 populations, a separate systematic review and meta-analysis focused on the younger age group may be warranted. When we report the results of this study, we will mention this in the limitations section and suggest the need for a future systematic review and meta-analysis focused on the under 40 population.

2. Vision impairment (VI) and blindness are two different states of visual dysfunction (see the definition by WHO), but mixed by the authors (Page 8 Lines 6-20). I suggest divide them into two parts, and analyze them respectively.

RESPONSE:

We appreciate this comment and agree that this approach makes sense and might provide additional valuable insights. Therefore, we have noted on Page 6 (line 154) that “People with each of the following categories of VI will be compared to those with better vision” and on Page 8 (line 224) that “We will extract definitions of VI and blindness to permit analyses based on specific levels of VI or blindness, insofar as there are sufficient data available to do so.”

3. For “Electronic searches”, no specific deadline was presented clearly, only “to present” (Page 9 Lines 7-13).

RESPONSE:

Thank you for noticing this. We have changed all of the places where we had stated “to present” to read “to 2019”.

4. The statistical methods used in the protocol were not described fully.

RESPONSE:

We have added some additional details to the statistical methods section to describe in more detail the specific statistical techniques that we will use. On Page 9 we have noted that “The log of each study estimate and its confidence intervals will be used to determine the study standard error; these will be then pooled using random-effects meta-analysis before taking the exponent of the results to present the pooled effect estimate on the original scale” and “The meta-regression outcome variable will be the log of the effect estimate for each study, and the aforementioned covariates will be included where data are available to do so.”

5. For some systemic diseases, such as diabetes, they both cause vision impairment and mortality. In this case, how to evaluate the association between vision impairment and mortality was not presented clearly in this protocol.

RESPONSE:

We agree that this is an important concern. This is the reason that we chose to exclude all studies that focused on specific populations (e.g. diabetes and stroke are common competing risk factors). We have also edited our conceptual framework in Figure 1 to explicitly point out that systemic health conditions like diabetes, hypertension, cardiovascular disease, and stroke are associated with both vision impairment and mortality. Based on this framework we are preferentially including effect estimates that adjust for likely confounders such as the presence of conditions like the aforementioned health conditions. Additionally, older age is likely the most common competing risk factor for mortality. Thus, we are only including studies that provide age-adjusted mortality estimates. Accordingly, we have noted on Page 6 (line 135), “Since age is a strong risk factor for mortality and VI, estimates of the effect of VI on mortality risk must be age-adjusted. Interventional studies and studies where all participants had a specific systemic disease (e.g. diabetes) will be excluded due to the difficulty of separating the possible effect of VI on mortality from the effect of an intervention or systemic disease on mortality.”

Reviewer: 3

Excellent. Important work. Well written.

Comments:

1. One of the potential limitations identified (p.3/17) relates to the "complexity of pathways between eye health and mortality (...) [being] unlikely to be fully described and tested in the current literature." Could you elaborate on how you plan to address this difficulty, and the impact this could have on the theoretical framework presented (p. 6, 15/17)?

RESPONSE:

Thank you for drawing attention to this interesting and important issue. A full discussion of the complex pathways that likely mediate the VI-mortality association is beyond the scope of this protocol paper. However, we have added a sentence to the introduction stating “Notably, the complex pathways that may mediate the association between VI and mortality may not have been fully described or tested in prior studies, though doing so will be an important future step toward optimizing outcomes for those with VI and blindness.” We will not be able to address this in our meta-analysis, other than by preferentially extracting effect measures that adjust for theoretical confounders but not mediators (as depicted in our theoretical framework); we have described this approach on Pages 7-8.

2. Sensitivity analysis (p. 12/17): "studies are excluded (...) if the risk of bias could not be adequately assessed". Would it be possible to describe (p. 10/17) the circumstances under which you expect this difficulty to occur?

RESPONSE:

We have removed this phrase since if data are not available to assess risk of bias in a given domain, this in fact characterizes the study’s “risk of bias”. This sentence now simply reads, “We will conduct a sensitivity analysis in which studies are excluded if they are judged to be at high risk of bias.”

Reviewer: 4

This systematic review and Meta-analysis has the potential to make an important contribution to global eye health. It is a well written protocol.

Some comments: 1) The authors have decided to exclude intervention studies where all participants had a specific systemic disease (e.g diabetes). 2) In addition, they have decided that men and women age 40 years and above at the time of the enrolment will be eligible for inclusion. I'm looking forward to read the strength and limitations (specific for 1 and 2) in the discussion chapter for the inclusion and the exclusion criteria.

RESPONSE:

Thank you for the kind comments. We have in fact adopted the aforementioned exclusion criteria, as described in the protocol. We assume that the reviewer is referring to looking forward to reading about these limitations when we publish final results of the systematic review/meta-analysis (as there is no discussion section in protocol papers). We plan to include discussion of these issues when we prepare the paper to disseminate our final results and look forward to sharing this with the scientific community.

Reviewer: 5

A very well planned study using established, published methodology. The results will be be very meaningful for advancing the important of visual function in older adults.

RESPONSE:

Thank you for the comments. We look forward to sharing the results of our final study.

VERSION 2 – REVIEW

REVIEWER	Marie-Josée Aubin Université de Montréal, Canada
REVIEW RETURNED	15-May-2020
GENERAL COMMENTS	Excellent. Thank you for addressing the comments in the revised version.

	I bring to your attention some typo errors in the abstract (one should read "PRISMA") and on p. 7/19, line 82 (rather "than" - word missing). On p. 9/10 line 201, please describe how disagreement between the "two reviewers [who] will independently assess the risk of bias" would be resolved. Consider adding the same sentence as in the section above : "Disagreements will be resolved by discussion and consultation with another author as needed." As appropriately anticipated, the heterogeneity in the measurement of visual acuity and in the definition of visual impairment (VI) might pose a significant challenge when comparing studies. Another comment relates to the difficulty in disentangling the pathways that link (directly or indirectly) VI to mortality. Notwithstanding these challenges and limitations, I am looking forward to this important comprehensive review.
--	---